# Gas-Phase vs. Grain-Surface Formation of Interstellar Complex Organic Molecules: A Comprehensive Quantum-Chemical Study

**DOI:** 10.3390/ijms242316824

**Published:** 2023-11-27

**Authors:** Berta Martínez-Bachs, Albert Rimola

**Affiliations:** Departament de Química, Universitat Autònoma de Barcelona, 08193 Bellaterra, Catalonia, Spain; berta.martinez@uab.cat

**Keywords:** astrochemistry, interstellar grains, water ice surfaces, molecule formation, iCOMs, computational chemistry

## Abstract

Several organic chemical compounds (the so-called interstellar complex organic molecules, iCOMs) have been identified in the interstellar medium (ISM). Examples of iCOMs are formamide (HCONH_2_), acetaldehyde (CH_3_CHO), methyl formate (CH_3_OCHO), or formic acid (HCOOH). iCOMs can serve as precursors of other organic molecules of enhanced complexity, and hence they are key species in chemical evolution in the ISM. The formation of iCOMs is still a subject of a vivid debate, in which gas-phase or grain-surface syntheses have been postulated. In this study, we investigate the grain-surface-formation pathways for the four above-mentioned iCOMs by transferring their primary gas-phase synthetic routes onto water ice surfaces. Our objective is twofold: (i) to identify potential grain-surface-reaction mechanisms leading to the formation of these iCOMs, and (ii) to decipher either parallelisms or disparities between the gas-phase and the grain-surface reactions. Results obtained indicate that the presence of the icy surface modifies the energetic features of the reactions compared to the gas-phase scenario, by increasing some of the energy barriers. Therefore, the investigated gas-phase mechanisms seem unlikely to occur on the icy grains, highlighting the distinctiveness between the gas-phase and the grain-surface chemistry.

## 1. Introduction

The interstellar medium (ISM) is the region between stars. Although not visible to the naked eye, ISM contains matter in the form of gas-phase molecules and solid-state dust grain particles (with a mass proportion of 99% and 1%, respectively). In dense molecular clouds (regions where star formation begins and is featured to have temperatures of 5–10 K and gas density of 104 cm^−3^), interstellar grains are sub-micrometer-sized particles, constituted by a refractory core (of silicates or carbonaceous materials) that is covered by ice mantles, predominantly of H_2_O but also of other volatile species like CO, CO_2_, NH_3_, and CH_3_OH, all of them identified using near-infrared observational measurements [1,2]. Regarding the gas-phase components, to date, approximately 300 interstellar species have been identified via rotational emission spectroscopy in the millimeter and sub-millimeter range [3]. Among these species, ca. one-third belongs to the group called interstellar complex organic molecules (iCOMs), which are carbon-bearing species containing at least five atoms [4,5,6]. iCOMs can be precursors of more-complex organic species, in which some of them can be of biological interest [7,8,9,10]. Therefore, the study of iCOMs formation is important not only for understanding the chemical complexity of space but also for offering valuable insights on the potential origins of organic compounds that could be relevant to the chemistry of life.

Currently, there are two prevailing paradigms for explaining the formation of iCOMs in the ISM: either through gas-phase reactions or on the surface of interstellar grains [11,12,13,14,15,16]. In both theories, the first step advocates the presence of hydrogenated species in the icy grains (e.g., H_2_O, NH_3_, CH_4_, H_2_CO, or CH_3_OH), which are formed by hydrogenation (i.e., H addition) onto atoms and simple species present on the grains [17]. For instance, H addition to O, N, and CO can give rise to the formation of H_2_O, NH_3,_ and CH_3_OH, respectively [18,19,20].

In the gas-phase scenario, the frozen hydrogenated species partially desorb into a gas-phase through thermal or non-thermal processes, where they react with other gaseous species forming iCOMs [13,14,21,22]. Due to the extremely low temperatures, feasible pathways in these harsh environments require both exothermic reactions and barrierless mechanisms or, at least, pathways presenting low energy barriers. Therefore, gas-phase reactions involve either ions or radicals, and can be categorized into two main groups: ion-neutral or neutral–neutral reactions. Regarding the neutral–neutral reactions, they are reactions between a radical and a closed-shell species, and those that are not barrierless are characterized to form a pre-reactive complex [16]. At very low temperatures, the prereactive complex can remain stable long enough to allow the system to overcome the low energy barrier of the reaction [23]. Neutral–neutral reactions primarily lead to the formation of a radical intermediate (because of the reactive coupling of the two species), which, under strict gas-phase conditions, cannot dissipate its internal energy, and accordingly cannot be stabilized. Due to its high internal energy, this radical intermediate can either dissociate back or evolve, yielding newly formed species [14,21,22,24,25].

In the grain-surface scenario, the hydrogenated icy species trapped on the grain surfaces are photo-dissociated by ultra-violet (UV) photons generating radicals. These radicals become more mobile when the grain temperature increases to about 20–30 K during the collapse phase, allowing them to diffuse over the surface and recombine to form iCOMs, in which such a radical–radical coupling is assumed to be barrierless [11,12]. Finally, iCOMs are released to the gas phase using thermal or non-thermal processes. However, calculations have shown that radical–radical reactions on water ice surfaces can present barriers and can have competitive channels, such as H-abstraction, significantly limiting the efficiency of the processes [26,27,28,29,30]. Therefore, alternative processes towards iCOMs synthesis have started to be considered, such as non-diffusive processes [31,32] involving gas–grain reactions between radicals (gas) and components of the ices (grain) [27,33,34,35].

The mechanisms and pathways that explain the formation of iCOMs still constitute a vivid debate. However, it seems that both paradigms, the gas-phase and the grain-surface mechanisms, are essential to account for the diversity and abundances of iCOMs observed in the ISM [11]. Determining whether the formation of a particular iCOM is governed by the gas-phase or the grain-surface chemistry is a complex matter since it depends on the intrinsic nature of the target iCOM, and accordingly a case-by-case investigation is required [16].

In this work, we investigate, by means of quantum-chemical simulations, the formation pathways of four astrochemically relevant iCOMs that are recurrently identified in different star-forming regions: formamide (HCONH_2_), acetaldehyde (CH_3_CHO), methyl formate (HCOCH_3_), and formic acid (HCOOH). Our approach involves a comprehensive investigation of the formation routes of these species on interstellar water ice models. However, the mechanisms under study in this contribution are inspired by the corresponding neutral–neutral gas-phase synthetic routes, which are considered to be major sources of these species in the gas phase. These gas-phase formation routes have not been considered on the ice mantles, and intriguingly, the interaction of the reactive species with the surfaces can modify the energetic features of the reactions, either increasing or decreasing the energy barriers, and favoring or disfavoring their thermodynamics. Accordingly, our goal is twofold: (i) to assess whether these gas-phase formation routes can also be potential paths for their synthesis on interstellar water ice mantles, and (ii) to determine any analogy or lack thereof of the gas-phase mechanisms compared to reactions occurring on the surfaces of interstellar water ice mantles.

## 2. Results

### 2.1. Gas-Phase Reactions

Initially, the reactions were simulated under gas-phase conditions (namely, in the absence of the water ice models) using the CCSD(T)//DFT methodology (see Section 4.2). These primary calculations, in addition to allowing us to understand the intrinsic electronic properties of the reactions, also serve to carry out a benchmarking study to select the most accurate DFT functional that best describes each of them. Details of the benchmarking study can be found in Appendix A. It is worth noting that in the energetics of these gas-phase reactions, the ZPE-corrections added to the CCSD(T) potential-energy values are those computed at the DFT/6-311+G(d,p) theory level.

#### 2.1.1. Formamide Formation

Formamide (HCONH_2_) is an iCOM that was first detected in the ISM in 1971 [36]. Since then, it has been identified in several star-forming regions, becoming a species commonly found in such regions [37,38,39,40,41,42,43]. Formamide is an iCOM of great interest because it is the simplest iCOM containing the four most important elements from a biological point of view: H, C, N, and O. Furthermore, formamide is the simplest molecule containing an amide bond (-CO-NH-), which is the same bond that joins amino acids in peptides. Therefore, formamide represents a turn out in chemical complexity within the ISM, increasing its interest from the standpoint of prebiotic chemistry.

Several pathways relative to the formation of formamide on grain surfaces have been proposed [27,29,44,45], but recently, a neutral–neutral gas-phase formation route for formamide has been postulated by Barone et al. to be particularly favorable [24]. It consists of a two-step reaction: the first step involving the coupling of formaldehyde (H_2_CO) and the NH_2_ radical, leading to the formation of a radical intermediate (H_2_CONH_2_), and the second step involving the dissociation of the radical intermediate yielding formamide and a hydrogen atom. That is,
H_2_CO + NH_2_ → H_2_CONH_2_ → HCONH_2_ + H(1)

This mechanism, although the PES presents potential energy barriers, was considered to be barrierless because most of the transition states are lower in energy than the initial reactants. In this mechanism, the radical H_2_CONH_2_ intermediate was found to be the most stable species on the PES, and, thus, the products were less stable than the intermediate. However, after adding the ZPE corrections, the intermediate and the product exhibit similar stability. In some studies, the product is the most stable structure [24], while in others the intermediate is the most stable one [25,46]. Since this reaction is associated with the release of an H atom, kinetic calculations based on the Rice–Ramsperger–Kassel–Marcus (RRKM) theory demonstrated its feasibility in the ISM [25,46].

The formation of formamide following this Reaction (1) is characterized here as CCSD(T)//DFT, in which the M06-2X functional is found as the most accurate DFT method for this reaction (see Appendix A for details). Figure 1 displays the geometries and the ZPE-corrected PES for this reaction. The results obtained agree with the previous studies, identifying the intermediate radical I as the most stable structure of the pure PES. However, similar to previous studies [24,25,46], after adding the ZPE-corrections, the intermediate and the products exhibit similar stability, the product being 3.7 kJ mol^−1^ more stable than the intermediate. The first step of the reaction (the coupling between the reactants to form I) exhibits an intrinsic energy barrier (energy difference between TS1 and PR, see Figure 2) of 22.6 kJ mol^−1^, in which the TS1 is 18.8 kJ mol^−1^ higher than the reactants R asymptote. The second step (intermediate dissociation) presents an intrinsic energy barrier (here, as energy difference between TS2 and I) of 45.6 kJ mol^−1^, in which the TS2 is 11.2 kJ mol^−1^ higher than the reactants R asymptote.

#### 2.1.2. Acetaldehyde Formation

Acetaldehyde (CH_3_CHO) is one of the most common iCOMs identified in the ISM, and first detected in 1971 [47]. Furthermore, acetaldehyde is a possible precursor of several carbohydrates, thus being an iCOM of great prebiotic potential [48,49]. Different acetaldehyde formation routes have been discussed in the literature, including both gas-phase synthesis and reactions occurring on the surfaces of water ice mantles [26,28,35,45,50,51,52,53,54]. Skouteris et al. [21] proposed a neutral–neutral gas-phase mechanism based on the ethanol tree, in which ethanol (CH_3_CH_2_OH) is considered the parent molecule of different iCOMs such as glycolaldehyde (HCOCH_2_OH) formic acid (HCOOH), acetic acid (CH_2_COOH), and also acetaldehyde.

The formation of acetaldehyde from the ethanol tree mechanism involves two sequential reactions. The first reaction is the α H-abstraction from ethanol using an OH radical, resulting in the formation of the CH_3_CHOH radical and water,
CH_3_CH_2_OH + OH → CH_3_CHOH + H_2_O(2)

The H-abstraction reaction can occur at three different sites, resulting in the formation of the radicals CH_2_CH_3_OH, CH_3_CHOH, or CH_3_CH_3_O, and water. Theoretical studies investigating this reaction, either identified the α H-abstraction transition state slightly below the energy level of the reactants (2.5 kJ mol^−1^) [55] or slightly above (0.7 to 1.8 kJ mol^−1^) [56,57]. Additionally, several experimental investigations, conducted at temperatures relevant to atmospheric or combustion conditions, have concluded that the α H-abstraction is the primary channel for this reaction [58,59].

The second reaction is the addition of an oxygen atom to the CH_3_CHOH radical, resulting in the formation of a radical intermediate CH_3_CHOOH, which subsequently dissociates to yield acetaldehyde and an OH radical,
CH_3_CHOH + O → CH_3_CHOOH → CH_3_CHO + OH(3)

This second reaction was theoretically investigated by Skouteris et al. [21]. The first step of this second reaction was found to be truly barrierless, in which the coupling of the atomic O(^3^P) to the CH_3_CHOH radical occurred in a spontaneous way, forming directly the CH_3_CHOOH intermediate. The second step (intermediate dissociation) presented an energy barrier but submerged below the energy of the reactants. The CH_3_CHOOH intermediate was found to be the most stable structure of the PES, giving rise to overall unfavorable reaction energetics. Additionally, the branching ratio for the acetaldehyde formation compared to other possible channels (i.e., formic acid and acetic acid formation), was found to be only 0.075. However, Vazart et al. [22] by means of RRKM-based kinetic calculations estimated the overall rate constant for the acetaldehyde formation, suggesting that this ethanol-tree-based path is kinetically favorable as a potential gas-phase source for the formation of interstellar acetaldehyde.

Here, we calculate this gas-phase formation route for acetaldehyde at the CCSD(T)//M06-250 2X-D3 level (the most accurate methodology among the tested ones, see Appendix A for details). The geometries and the ZPE-corrected PESs are illustrated in Figure 2. For Reaction (2), the results identify the products P1 as the most stable structure of the profile. Although this reaction exhibits an intrinsic energy barrier of 18.1 kJ mol^−1^ (energy difference between TS1 and PR), the transition state TS1 is slightly below in energy than the reactant R1 asymptote (−1.3 kJ mol^−1^). In contrast, for Reaction (3), the intermediate I is the most stable structure along the pathway, to the detriment of the final product P2. The O(^3^P) addition step is barrierless, as it is a radical–radical coupling, while the dissociation step of I presents an intrinsic energy barrier of 72.8 kJ mol^−1^ (energy difference between TS2 and I). However, it is noteworthy that the energy level of the transition state TS2 is significantly lower than that of the reactants R2 asymptote.

**Figure 2 ijms-24-16824-f002:**
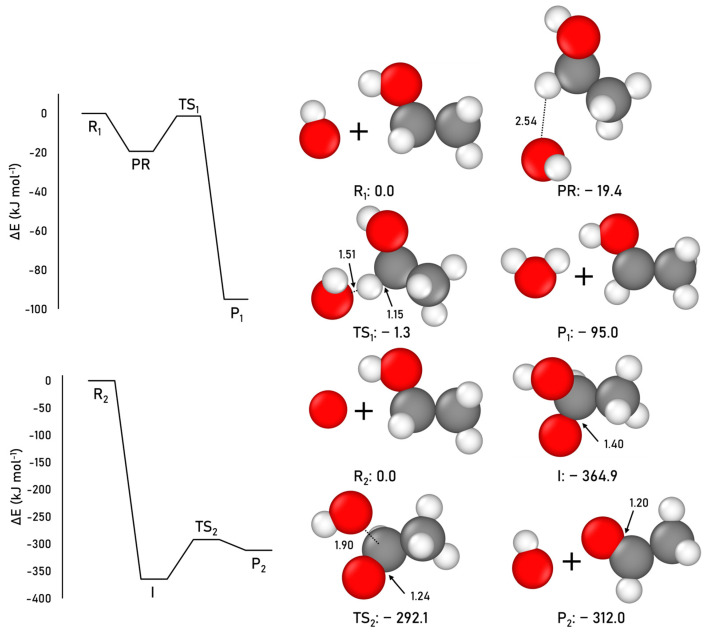
CCSD(T)//M06-2X-D3 ZPE-corrected PES and relative energies (with respect to the corresponding R1 and R2 asymptotes), and M06-2X-D3-optimized geometries for the gas-phase acetaldehyde formation. Distances are in Å and relative potential energies in kJ mol^−1^. Color coding: white, H atoms; grey, C atoms; and red, O atoms.

#### 2.1.3. Methyl Formate Formation

Methyl formate (HCOOCH_3_), the simplest ester identified in the ISM, was first detected in 1975 [60]. Moreover, methyl formate is detected in more abundance than its structural isomers acetic acid (CH_3_COOH) and glycolaldehyde (HCOCH_2_OH) [61].

Various mechanisms for the formation of methyl formate have been studied in recent years [11,62,63]. A neutral–neutral gas-phase synthetic pathway was proposed by Ruaud [64] and Balucani [14], pointing out dimethyl ether (CH_3_OCH_3_) as a gas-phase precursor for methyl formate. The proposed mechanism, previously investigated by Hoyermann [65] and Song [66], involves a two-step reaction. The first step is the addition of an oxygen atom to the CH_3_OCH_2_ radical, which is barrierless due to its radical–radical coupling nature, resulting in the formation of H_2_COOCH_3_ as a radical intermediate. The second step consists of the dissociation of this radical intermediate yielding methyl formate and a hydrogen atom,
CH_2_OCH_3_ + O → H_2_COOCH_3_ → HCOOCH_3_ + H(4)

Dedicated calculations found the radical intermediate as the most stable species of the path. Additionally, different competitive channels were explored, including the dissociation of the radical intermediate into several products such as CH_3_O and CH_2_O, CH_2_OH and CH_2_O, CH_3_ and HCOOH, HCO and CH_3_OH, and H_2_ and CH_3_CO. However, kinetic calculations indicated that the formation of methyl formate dominates over the other potential competitive channels, in which tunneling effects were underlined since the reaction involves the release of a hydrogen atom [66].

The synthesis of methyl formate following this neutral–neutral gas-phase mechanism is here investigated, in this case using the CCSD(T)//B3LYP-D3(BJ) methodology (for this reaction, the B3LYP-D3(BJ) method was found to be the most accurate, see Appendix A for details). Figure 3 shows the optimized geometries and the corresponding ZPE-corrected PES.

Our results are consistent with the previous studies. The radical intermediate is the most stable structure of the energy profile (more than the products P), the atomic O addition step is barrierless, and the intermediate dissociation step presents an intrinsic energy barrier of 49.0 kJ mol^−1^ (energy difference between TS and I). Nevertheless, the transition state TS is significantly more submerged in energy than the reactants R asymptote.

#### 2.1.4. Formic Acid Formation

Formic acid (HCOOH) is an organic molecule detected for the first time in the ISM in 1971 [67]. Formic acid is the simplest carboxylic acid and serves as a fundamental compound for the formation of more-complex carboxylic acids, including biomolecules [68]. Several pathways for the formation of formic acid have been considered in the literature [11,14,64,69,70]. The UMIST [71] and KIDA [72] databases report various methods of gas-phase synthesis for formic acid. One of them is the neutral–neutral gas-phase formation route, which is analogous to the two-step formamide formation route considered in this work. The first step involves the coupling between formaldehyde (H_2_CO) and a OH radical, leading to the formation of a radical intermediate, H_2_COOH. The second step consists of the dissociation of the radical intermediate, forming formic acid and releasing a hydrogen atom,
H_2_CO + OH → H_2_COOH → HCOOH + H(5)

Yetter [73] conducted experimentally studies on this reaction, although not under ISM conditions. They found that at temperatures above 200 K, a competitive channel of this mechanism, the H-abstraction leading to HCO and H_2_O, was the main branching ratio. D’Anna [74] performed a computational work on this reaction obtaining an energy barrier for the first transition state of 27.8 kJ mol^−1^; although, they did not consider the formation of the pre-reactant complex. Finally, Xu [75] focused on the simulation of the first step of this reaction, finding an energy barrier of 23.8 kJ mol^−1^. Despite these mixed results, different models consider this reaction as one of the major channels towards formic acid formation [11,12,76].

Here, we explore this formation pathway at the CCSD(T)//B3LYP-D3(BJ) level, in which for this reaction B3LYP-D3(BJ) was found to be the most accurate method (see Appendix A for details). The obtained results are shown in Figure 4.

The results are in very good agreement with the previous works. The most stable structure of the energy profile is the radical intermediate, I, which is significantly lower in energy than the final products P. The coupling step of the reaction presents an intrinsic potential energy barrier (energy difference between TS1 and PR) of 33.1 kJ mol^−1^ (very similar to that reported in previous works), in which the TS1 is 18.7 kJ mol^−1^ higher than the reactants R asymptote. The dissociation step presents an intrinsic energy barrier of 59.2 kJ mol^−1^ (energy difference between TS2 and I), but the TS2 structure lays below in energy with respect to the reactants R asymptote.

### 2.2. Reactivity on the Water Ice Surface Models

Aiming to determine if the previously studied gas-phase reactions can also occur on the surfaces of water ice mantles, we computed them on our CWS and ASW periodic water ice models. It is worth mentioning that on the CWS, due to its inherent crystallinity, only one surface reaction site is available. However, this is not the case for the ASW model, in which different adsorption sites can be identified, which are potential reaction sites. Nevertheless, computing all the reactions on all these surface sites is computationally overwhelming and in practice unfeasible. Here, we opted to simulate the reactions on one reaction site, which is the one present in the cavity of the ASW model. This choice is because interstellar ices are materials that seem to present some degree of porosity [1,77]. By proceeding in this way, our simulations align with a more realistic adsorption scenario from an interstellar ice perspective, while significantly reducing the potential for diverse adsorption sites (and the computational cost).

As mentioned in Section 4.2, for the calculations with these periodic systems, we adopted the cost-effective DFT//HF-3c approach to simulate the reactions, in which the DFT functional is that chosen in the benchmarking study completed in the gas phase (see Appendix A for more details on the choice of the functional). For all the reactions, the first step was the adsorption of the reactants by placing them manually on the surface.

#### 2.2.1. Formamide Formation

The formation pathway towards formamide (see Reaction (1)) was simulated on the CWS and ASW models, adopting the M06-2X-D3//HF-3c methodology. Figure 5 displays the optimized geometries of the stationary points of the reaction and the ZPE-corrected PES simulated on the two surface models.

The thermodynamics of the reaction remain consistent with those observed in the gas phase. The product, which includes HCONH_2_ and H adsorbed on the surface, represents the most stable stationary point of the energy profiles. However, it is noteworthy that the difference in relative energies between the product and the intermediate is more pronounced in both surface models (18.2 kJ mol^−1^ for CSW and 8.3 kJ mol^−1^ for ASW) than in the gas-phase results (3.7 kJ mol^−1^). This is because they are stabilized by favorable interactions with the surface, such as H-bonds and dispersion forces, which are absent in the gas-phase scenario.

Regarding the first step of the reaction (the coupling between H_2_CO and NH_2_), it presents very high energy barriers (63.7 and 51.7 kJ mol^−1^ on CWS and ASW, respectively), which contrasts to that in the gas phase (22.6 kJ mol^−1^), probably due to the need to break the reactant/surface interactions to reach the transition structure (which is not necessary in the gas phase). The difference between the energy barriers when on CWS or ASW models is probably due to the high rigidity of the CSW model, which inhibits stabilizing surface reconstructions that are possible in the softer ASW model. The second step (dissociation of the intermediate), in contrast to the first step, presents energy barriers of 35.6 and 38.3 kJ mol^−1^ on CWS and ASW, which are significantly lower than that in the gas phase (45.4 kJ mol^−1^).

#### 2.2.2. Acetaldehyde Formation

The acetaldehyde formation pathway (Reactions (2) and (3)) was simulated on the water ice models through employing the BHLYP-D3(BJ)//HF-3c methodology. In the benchmark study, the M06-2X-D3 functional provided the most accurate description of these reactions, but we encountered dramatic energy convergence problems in the single-point calculations within the periodic approach. Consequently, we moved on the BHLYP-D3(BJ) method, as it was the second most accurate method describing the reactions. The geometries of the stationary points of the reactions and the ZPE-corrected PES simulated on the CWS and ASW models are depicted in Figure 6.

Regarding the first reaction, the products formed on both surfaces (CH_3_CHOH + OH) constitute the most stable stationary point of the profile, as also found in the gas-phase scenario. In contrast, the identified energy barriers for this reaction are 5.6 kJ mol^−1^ on the CWS and barrierless on the ASW (in this latter case when ZPE-corrections are considered), which are significantly lower than that in the gas phase (18.1 kJ mol^−1^). As far as the second reaction is concerned, the results obtained on the CWS and ASW follow the same trend as in the gas phase, that is, a barrierless O addition to CH_3_CHOH forming the CH_3_CHOOH intermediate, which is the most stable structure of the energy profile, meaning that the final acetaldehyde product is thermodynamically disfavored because it is less stable than the intermediate. Moreover, the dissociation step of the intermediate to form acetaldehyde presents energy barriers of 341.8 and 217.8 kJ mol^−1^ on CWS and ASW, respectively, which are notably higher than that computed in the gas phase (72.8 kJ mol^−1^). The energy barrier difference between the CWS and the ASW can be attributed to the highest rigidity of the CWS that hampers the stabilizing surface reconstruction effects that are possible to take place in the less-rigid ASW system.

#### 2.2.3. Methyl Formate Formation

The formation of methyl formate adopting the Reaction (4) was simulated on the CWS and ASW models at the B3LYP-D3(BJ)//HF-3c theory level. Figure 7 displays the geometries of the stationary points of the reaction and the ZPE-corrected PES simulated on the two water ice surface models.

On the water ice models, the O addition to the CH_2_OCH_3_ radical to form the H_2_COOCH_3_ intermediate is also barrierless, as occurred in the gas phase, due to being a spin–spin-coupling driven reaction. Dissociation of this intermediate gives rise to the final reaction.

Methyl formate formation pathway presents an energy barrier of 32.5 kJ mol^−1^ on the CWS model, which is significantly lower than in the gas phase (49.0 kJ mol^−1^). However, on the ASW, this step presents an energy barrier as high as 97.9 kJ mol^−1^ (higher than in the gas phase). The elevated energy barrier in the ASW model is attributed to a substantial internal structural reorganization of the surface during the dissociation process, giving rise to this elevated energy barrier, while on the CWS model, the dissociation step occurs without significant structural reorganization.

Regarding the stability of the final products, on both CWS and the ASW, they present a similar stability as the intermediate; on the CWS the products are just 3.3 kJ mol^−1^ higher in energy, while on the ASW model they are just 2.0 kJ mol^−1^ lower in energy. This trend is analogous to the gas-phase reaction, where both the intermediate and the products display similar stability, the intermediate being the most stable structure of the profile and 6.7 kJ mol^−1^ more stable than the products.

#### 2.2.4. Formic Acid Formation

The formation of formic acid, through the Reaction (5), was simulated on the CWS and ASW models using the B3LYP-D3(BJ)//HF-3c approach. The ZPE-corrected PES and the geometries of the stationary points involved in this formation route on the two water ice models are depicted in Figure 8.

The results obtained on CWS and ASW indicate that the presence of the water ices reverses the energetic trend with respect to the gas-phase scenario. That is, on the icy surfaces, the final products (HCOOH and H on the surfaces) are the most stable stationary points of the path. This was also observed in the formation of formamide on the water ices (see above) and can be attributed to the emergence of favorable interactions between the newly formed products and the surface, such as H-bond and dispersion, which are missing in gas-phase conditions.

The coupling step presents high energy barriers on the water ice surface models (59.5 and 83.6 kJ mol^−1^ on CWS and ASW, respectively), which are significantly higher energy barriers than those in the gas phase (33.1 kJ mol^−1^), due to the need to break the reactants/surface interactions. In contrast, the dissociation step presents energy barriers of 23.7 and 22.1 kJ mol^−1^ on CWS and ASW, respectively, which are lower than those obtained in the gas phase (59.2 kJ mol^−1^).

## 3. Discussion

In this work, we modeled the formation pathways of four astrochemically relevant iCOMs: formamide, acetaldehyde, methyl formate, and formic acid. As formation routes, we took feasible gas-phase neutral–neutral reactions, which, additionally, were also simulated on models of CWS and ASW ices. Table 1 summarizes the energetics of the simulated reactions.

All these processes involve multi-step mechanisms. In the gas-phase scenario, the reaction intermediates are usually the most stable stationary points, and when they are not, are very similar in energy as the final products. This could lead us to think that the processes are thermodynamically disfavored, but (as mentioned in the Introduction), in the gas phase, whether a reaction is favorable or not is not only regulated by their energetic features, but overall kinetics need to be accounted for, normally by means of calculations based on the RRKM theory. In this sense, available studies indicate that these paths are actually favorable and hence they have been proposed to be major sources for these iCOMs. A particular feature enabling the occurrence of these reactions is that the energy of the transition states is lower than the asymptotes, meaning that the system has enough internal energy overcoming the energy barriers. On the other hand, when these reactions are simulated on the periodic water ice models, in some cases, the results indicate that the presence of the surface alters the energetic trends compared to the gas-phase conditions. That is, in the formation of formamide and formic acid, the reaction products are identified as the most stable structures, more than the respective intermediates. In the formation of methyl formate, the stabilities between the products and the intermediate are very similar, while the formation of acetaldehyde is the unique case in which the intermediate is still the most favorable structure along the path. Modification of the thermodynamics of the reactions on the surface with respect to the gas-phase scenario is due to the interactions between the structures and the surface, which cannot take place in the gas phase. These interactions (mostly based on H-bonding and dispersion) have been found to be indeed important in driving reactivity of similar organic molecules, which are used as conformers in co-crystals in relevant processes for pharmaceutical industry [78,79,80,81].

Although all the reactions considered in this work are neutral–neutral reactions, to facilitate the discussion, we can categorize the steps studied as (i) H abstraction, which is the first reaction of the formation mechanism of acetaldehyde (Reaction (2)); (ii) addition of atomic O to a radical, the first step of the formation mechanism of methyl formate (Reaction (4)) and also the first step of the second reaction of the formation mechanism of acetaldehyde (Reaction (3)); (iii) coupling of formaldehyde with a radical, in particular with NH_2_ (first step of formamide formation, Reaction (1)) and with OH (first step of formic acid formation, Reaction (5); and (iv) dissociation of intermediates, which occurs in the second step of formamide, methyl formate, and formic acid formations (Reactions (1), (4), and (5), respectively) and also in the second step of the second reaction of acetaldehyde formation (Reaction (3)).

When considering the reaction belonging to the category (i)—H abstraction—in the gas phase, it exhibits a high intrinsic energy barrier (18.1 kJ mol^−1^). As mentioned above, despite this barrier, the TS is 1.3 kJ mol^−1^ below the asymptote (see Figure 2) and hence the reaction is assumed to be favorable under these conditions. When this process is simulated on the periodic water ice models, a transition state is identified on both models. On the CWS, the ZPE-corrected barrier is 5.6 kJ mol^−1^, while on the ASW, after adding the ZPE-corrections, the barrier is submerged below the reactants, and accordingly it can be considered barrierless. Irrespective of the model, it is clear that the presence of the water ice surface decreases the energy barriers, and this way exerting a catalytic effect. This is probably due to the interactions of the CH_2_ moiety of ethanol and the OH radical with the surface which polarize the C-H and O-H chemical bonds, facilitating the H abstraction. Therefore, considering that interstellar water ices are amorphous, we can state that these H abstraction processes are likely to occur in the ISM as a grain-surface reaction.

When considering the processes belonging to the category (ii)—O addition to radicals –either in the gas phase or on the water ice surfaces, they are barrierless. This is because the process is driven by a spin–spin coupling, which under gas-phase conditions are definitely barrierless, but it could not be the case on water ice surfaces. Indeed, other radical–radical processes (also driven by spin–spin couplings) have shown that these processes can present energy barriers, the emergence of which are primarily attributed to the interactions between the radicals and the surface (mainly H-bonds and dispersion), which need to disrupt so that the reactions proceed. In a study by some of us [29], it was established that the interactions with the surface play a crucial role in the magnitude of these energy barriers. Specifically, strong radical–surface interactions (like H-bonds) are expected to result in relatively high energy barriers because it is necessary to break them for the progress of the reaction, which is translated to an increased energy barrier. In contrast, for weak radical-surface interactions (like the dispersive forces), low energy barriers or even barrierless reactions are expected, because reaching the transition state structure involves essentially the rotation and/or translation of the reactive species, which are low energetic motions. In our specific cases, both the CH_3_CHOH and CH_3_OCH_2_ radicals are firmly attached to the water ice surfaces due to forming H-bonds. However, atomic O is weakly bounded to the surface (it indeed presents a very low binding energy, 13 kJ mol^−1^ [82], enabling an efficient diffusion on icy water surfaces [83,84]). In our simulations, we observed that only a slight rotation of the radical to become well oriented with the O atom was required to proceed with the coupling, which resulted in barrierless reactions. Therefore, these processes can also be considered as feasible in the ISM.

When considering the processes falling into category (iii)—formaldehyde–radical couplings—we observe an increase of the energy barriers when they occur on the water ice surfaces when compared with the gas-phase ones. This can be attributed to the favorable interactions between the reactants and the surface leading to an extra-stabilization of the reactants, which is not possible in the gas phase. However, we did not find a consistent trend between the energy barriers on the CWS and the ASW models. That is, in the formamide case the energy barrier is larger on CWS than on ASW, while in the formic acid case it is the opposite. As mentioned above, to enable the system to overcome the activation barriers, it is required to break the interactions between reactants and the surface, and thus stronger interactions with the surface are predicted to result in higher activation barriers. It is worth mentioning that in the works of Perrero et al. [85], in which the binding energies (BEs) of several species using the same periodic ice models as here were computed, it was found that a distribution of BEs could be determined when using the ASW model due to the presence of different binding sites, which are also different reaction sites. Therefore, depending on the initial positions of the reactants and their interactions with the surface, the energy barrier can either increase or decrease when transitioning from the CWS model to the ASW model. Irrespective of that, these formaldehyde–radical coupling processes present substantial energy barriers and accordingly they could hardly happen in interstellar conditions.

When considering processes of category (iv)—intermediate dissociation—they present substantial energy barriers when occurring both in the gas-phase conditions (considering intrinsic energy barriers) and on the water ice models. As mentioned above, neutral–neutral reactions usually lead to the formation of a radical intermediate, which under strict gas-phase conditions cannot be stabilized by dispersing its internal energy, and, thus, the reaction can dissociate back or evolve to form new species (in exothermic reactions). In contrast, when these reactions occur on water ice surfaces, the water ices can act as third bodies (as demonstrated by some of us in recent works [19,86,87]), in which the intermediate is stabilized by transferring the energy released by the reaction to the ice. Consequently, the intermediate becomes stable on the surface and does not contain the same excess of internal energy as occurs in the gas phase, and accordingly substantial dissociation energy barriers emerge, which are hard to surmount in ISM conditions. However, when the intermediate dissociation is associated with the release of a H atom (as is the case of formamide, methyl formate, and formic acid formation), tunneling effects can become relevant, especially at very low temperatures [88,89]. Tunneling is more likely in symmetric processes than in non-symmetric processes, that is, when the initial and final states present similar stabilities. This is the case of methyl formate on the water ices, in which the intermediate and its dissociated product are almost isoenergetic. To check upon the possibility of tunneling, we conducted a simplistic (but useful) assessment by calculating the crossover temperature (T_X_), namely, that temperature below which tunneling dominates, by adopting the Fermann Auerbach formulation [90], which accounts directly for the energy barrier height and indirectly for the energy barrier width with the transition frequency. The results show, for this process, T_X_ of 217 and 399 K on CWS and ASW, respectively, and accordingly tunneling can dominate for the interstellar conditions, making this step possibly favorable. Bearing in mind that the first step of the methyl formate formation is the O addition to CH_2_OCH_3_ (barrierless), we can state that its formation through Reaction (4) is plausible in the ISM. In the formation of formamide and formic acid, the intermediate dissociation is a non-symmetric process, in which the final product is more stable than the intermediate, which is also a favorable situation for the dominance of tunneling (but less than for symmetric barriers). To assess the tunneling effects on these two processes, we also calculated the T_X_, which are 233 and 233 K, and 223 and 257 K on CWS and ASW in the formamide and formic acid formation, respectively. Thus, the intermediate dissociation by releasing an H atom in the final step of the formamide and formic acid formation seems to be feasible under tunneling regimes in the ISM. Nevertheless, it is worth remembering that, at variance with the methyl formate formation, the initial steps in the formation routes of these two compounds have been found to be unlikely in the ISM due to presenting high energy barriers, so that the formation of these iCOMs is highly hampered. Despite this, it is worth mentioning that the presence of the water ice surfaces modifies the relative energies between the intermediates and the products, always in favor of the stabilization of the products. Thus, the icy surfaces allow the obtaining of energy barrier situations (namely, symmetric, and exoenergetic non-symmetric) favorable for the occurrence of tunneling. Obviously, a more rigorous treatment of the tunneling effects is needed to definitely determine their actual role in these steps, such as applying semi-classical approaches to the transition state and RRKM theories [91] or the instanton theory [92].

In summary, our results predict that gas-phase processes can hardly occur on water ice surfaces. The only reaction which can potentially occur is the formation of methyl formate, in which tunneling effects need to be advocated. Thus, a main message of this work is that there is a clear frontier between gas-phase and grain-surface chemistry, that is, gas-phase processes can be feasible as specifically gas-phase reactions but not as grain-surface reactions, and vice versa. Consequently, results emerging from one scenario cannot be directly translated to the other. Therefore, establishing a general trend for the lack of analogy between gas-phase and grain-surface mechanisms is challenging, as the feasibility of the reactions depends on the nature of the reaction steps, the reactive species involved, and the surface model, thus requiring a case-by-case investigation.

## 4. Materials and Methods

### 4.1. Surface Modelling

A periodic approach was adopted to model the surfaces mimicking the interstellar water ice mantles. Two different models were considered: one crystalline and another one amorphous.

The crystalline ice surface model was based on the P-ice structure. P-ice represents the proton ordered counterpart to the hexagonal (proton-disordered) water ice, which has been shown to replicate well the physico-chemical properties of crystalline water ice [93]. The 2D periodic surface model was generated by cutting the P-ice 3D periodic bulk structure perpendicular to the [010] direction. As a result, the (010) slab surface model was obtained, which is one of the most stable planes among crystalline P-ice surfaces, presenting a null dipole component perpendicular to the surface [94,95]. The resulting crystalline water surface model (hereafter referred to as CWS) contains 12 atomic layers, with a total thickness of 10.90 Å, and includes 24 water molecules (72 atoms) per unit cell.

However, the CWS model is not representative of the structural features of actual interstellar water ices, as they are predominantly found in an amorphous state [1,96]. To better depict a more realistic interstellar water ice surface, an amorphous model designed by Ferrero et al. [97] was also employed. The amorphous surface was constructed by assembling small clusters generated by Shimonishi et al. [98]. Like the CWS model, this amorphous solid water model (hereafter referred to as ASW) is based on a periodic approach but in order to account for the amorphousness of the system, a larger unit cell was used (20.35 Å of length), containing 60 water molecules (180 atoms) per unit cell.

Figure 9 displays the top and lateral view of CWS and the lateral view of ASW structural models.

### 4.2. Computatonal Details

Simulations were carried out employing the periodic ab initio CRYSTAL17 code [99]. This software implements both the Hartree–Fock and the Kohn–Sham self-consistent field methods to solve the electronic Schrödinger equation. CRYSTAL17 uses localized Gaussian type orbitals (GTOs) as basis sets, which are functions centered to the atoms that decay with the distance with respect to each nucleus, allowing accordingly the simulation of both periodic (crystals, surfaces, and polymers) and non-periodic (molecules). This is at variance with basis sets based on plane wave (PWs) functions, which are periodic functions that fill all the 3D space. Consequently, the definition of surfaces in CRYSTAL17 does not require the 3D replica commonly used in PW-based codes, and thus they are represented as true slab models, with a complete empty space above and below the slab.

The characterization of the potential energy surface (PES) of the reactions required determines the structures and the energetics of their stationary points, including reactants, intermediates, products, and transitions states. A cost-effective approach was employed for this purpose, referred to as DFT//HF-3c. In this composite procedure, the geometries of the stationary points were optimized at HF-3c level, and, subsequently, an energy refinement step was carried out through single-point energy calculations with a density functional theory (DFT) method on the optimized HF-3c geometries. HF-3c is a semi-empirical method within the Hartree–Fock framework, using a minimal Gaussian atomic orbital (Gaussian-AO) basis set known as MINIX. It includes three a posteriori corrections to alleviate the deficits arising from the approximations introduced. These corrections account for (i) short-ranged deficiencies due to the use of a small or minimal basis set; (ii) basis set superposition error (BSSE) implementing the geometrical counterpoise correction gCP); and (iii) dispersive interactions [100,101,102,103,104].

For the refinement of the energetics, different DFT functionals were considered: the pure gradient generalized approximation (GGA) PBE; the non-local hybrids B3LYP and BHLYP, which include a 20% and a 50% of exact exchange in its definition, respectively; and the meta-hybrid M06-2X, which incorporates a 45% of exact exchange [105,106,107]. The basic set used for these DFT calculations was the 6-311+G(d,p) one. Furthermore, to account for dispersive interactions, the corresponding Grimme’s D3 correction within the Becke–Johnson scheme (i.e., the D3(BJ) correction) was included when available; if not available, the D3 correction was used [108,109]. For both CWS and ASW, the Hamiltonian matrix was diagonalized in 4k points in the first Brillouin zone, corresponding to a shrinking factor of 2. Values of 7, 7, 7, 7, 14 for the tolerances controlling Coulomb and the exchange series were adopted for all calculations. The choice of the numerical values controlling these computational parameters is fully justified in Ferrero et al. [97]. For geometry optimization, the convergence criterium was set to 10^−7^ Hartree, meaning that the self-consistent fields (SCF) procedure was stopped when the energy difference between cycles was smaller than this value.

To identify the DFT functional that described more accurately each chemical reaction, a gas-phase benchmarking study was conducted using the molecular GAUSSIAN16 program [110]. This study involved optimizing geometries using the four different DFT functionals (PBE, B3LYP, BHLYP, M06-2X) with a 6-311+G(d,p) basis set and the energies refined by performing single-point energy calculations at the same DFT methods but employing a 6-311++G(2 df,2 pd) basis set. To assess the accuracy of the DFT methods, the results were confronted with those obtained at the CCSD(T) level (considered the “gold-standard” method of quantum chemistry), which were obtained by performing single-point energy calculations on the DFT-optimized geometries using the Dunning aug-cc-pVTZ basis set [111] (hereafter referred to as CCSD(T)//DFT methodology). By comparing the energy results computed at full DFT level with the CCSD(T)//DFT ones, the most suitable DFT functional for each reaction was selected (see Table A1). Moreover, since periodic simulations were based on a DFT//HF-3c scheme, we performed an additional benchmarking for the formamide gas-phase formation comparing this methodology with the CCSD(T)//DFT and CCSD(T)//HF-3c ones. The obtained results (shown in Table A1 and Table A2) show very good agreement between these methodologies, pointing out the reliability of the DFT//HF-3c scheme for the periodic systems.

To prevent surface deformations, periodic geometry optimizations were carried out by relaxing the internal atomic positions but by keeping fixed to the lattice parameters. Additionally, the spin-unrestricted formalism was adopted for open-shell systems. The nature of the stationary points of the reactions was characterized by calculating the harmonic frequencies at the HF-3c level; the outcome being minima for reactants, intermediates, and products, and first-order saddle points for transitions states. In CRYSTAL17, harmonic frequency calculation is conducted numerically at the Γ point by diagonalizing the mass-weighted Hessian matrix of the second-order energy derivatives with respect to atom displacements (i.e., central difference formula), in this case ±0.003 Å from the minimum along each Cartesian coordinate [112,113]. Here, in frequency calculations, the convergence criterium was set to 10^−10^ Hartree. With the harmonic frequency calculations, the vibrational zero-point energy corrections (ZPE) were obtained and applied to the corresponding potential-energy values.

## 5. Conclusions

In this study, we investigated the means of quantum-chemical simulations the formation pathways of four astrochemically relevant iCOMs: formamide, acetaldehyde, methyl formate, and formic acid. We explored these pathways by adopting neutral–neutral gas-phase synthetic routes (considered as major production sources of these molecules in the gas phase) and exporting them on water ice surface models. Our approach involved characterizing the potential energy surfaces for these mechanisms in the gas phase and on the periodic water ice models (crystalline and amorphous ones). We adopted a composite procedure, the DFT//HF-3c approach, in which we optimized geometries at the HF-3c level and then performed single-point energy calculations at the DFT level on the optimized HF-3c geometries. The DFT methods were benchmarked against CCSD(T) calculations in the gas-phase reactions.

The results obtained indicate that on the water ice models, while some reactions exhibit lower intrinsic energy barriers compared to the gas-phase results, or even barrierless situations, each mechanism investigated included at least one high energy barrier step, which at low temperatures are unlikely to be overcome, thus hampering the formation of these molecules on grain surfaces. The presence of such high energy barriers is due to the third body effect of the icy surfaces, which by dissipating the internal energies of the intermediates stabilize them. The only exception in which the iCOM can be formed adopting a gas-phase neutral–neutral mechanism on the water ice surfaces is through methyl formate, in which the energetic step can be dominated by tunneling effects.

An interesting outcome emerging from this work is that the presence of the surface can modify the overall energetics of the reactions compared with the gas-phase processes. While in this latter scenario, reaction intermediates are usually found to be more stable than the final products, on the water ice surface models this is not the case, in which the final products are in general more stable than the intermediates, thanks to the favorable interactions with the surface.

In conclusion, this study highlights the fundamental differences between gas-phase and grain-surface chemistry. Formation routes cannot be translated between these scenarios due to the modifying influence of the surface in the energy trends of the reactions. Accordingly, establishing a generalization between gas-phase and grain-surface chemistry is challenging, and, therefore, we advocate that a case-by-case study is required.

## Figures and Tables

**Figure 1 ijms-24-16824-f001:**
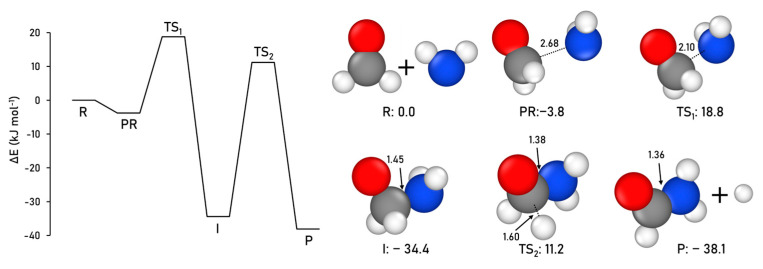
CCSD(T)//M06-2X-D3 ZPE-corrected PES and relative energies (with respect to the R asymptote), and M06-2X-D3-optimized geometries for the gas-phase formamide formation. Distances are in Å and relative potential energies in kJ mol^−1^. Color coding: white, H atoms; grey, C atoms; blue, N atoms; and red, O atoms.

**Figure 3 ijms-24-16824-f003:**
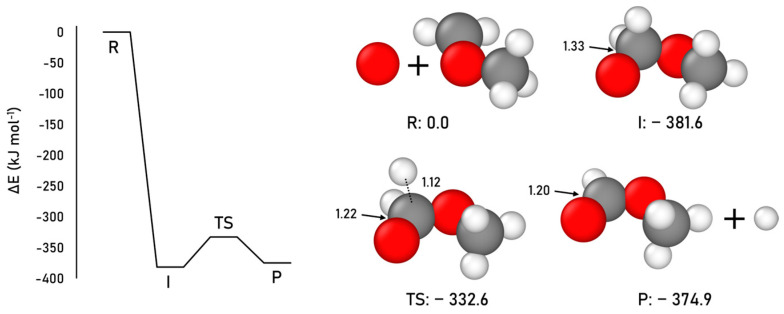
CCSD(T)//B3LYP-D3(BJ) ZPE-corrected PES and relative energies (with respect to the R asymptote), and B3LYP-D3(BJ)-optimized geometries for the gas-phase methyl formate formation. Distances are in Å and relative potential energies in kJ mol^−1^. Color coding: white, H atoms; grey, C atoms; and red, O atoms.

**Figure 4 ijms-24-16824-f004:**
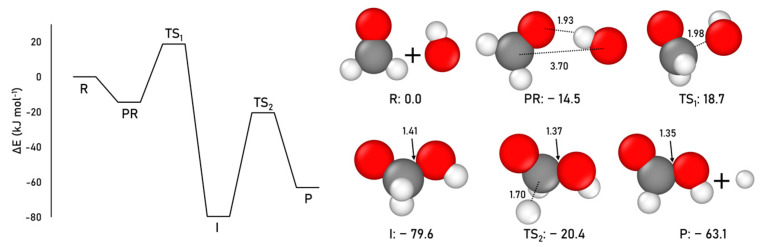
CCSD(T)//B3LYP-D3(BJ) ZPE-corrected PES and relative energies (with respect to the R asymptote), and B3LYP-D3(BJ)-optimized geometries for the gas-phase formic acid formation. Distances are in Å and relative potential energies in kJ mol^−1^. Color coding: white, H atoms; grey, C atoms; and red, O atoms.

**Figure 5 ijms-24-16824-f005:**
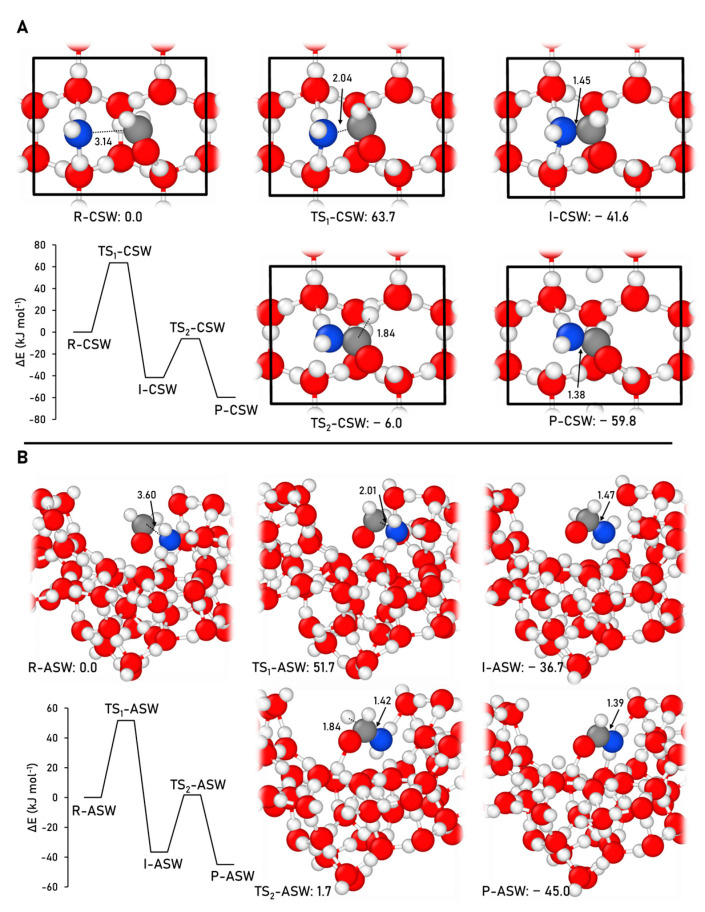
M06-2X-D3//HF-3c ZPE-corrected PES and the corresponding optimized stationary points, alongside the relative energies, for the formamide formation on the CWS (**A**) and ASW (**B**) models. The unit cell is highlighted in black in the CWS model. Distances are in Å and relative potential energies in kJ mol^−1^. Color coding: white, H atoms; grey, C atoms; blue, N atoms; and red, O atoms.

**Figure 6 ijms-24-16824-f006:**
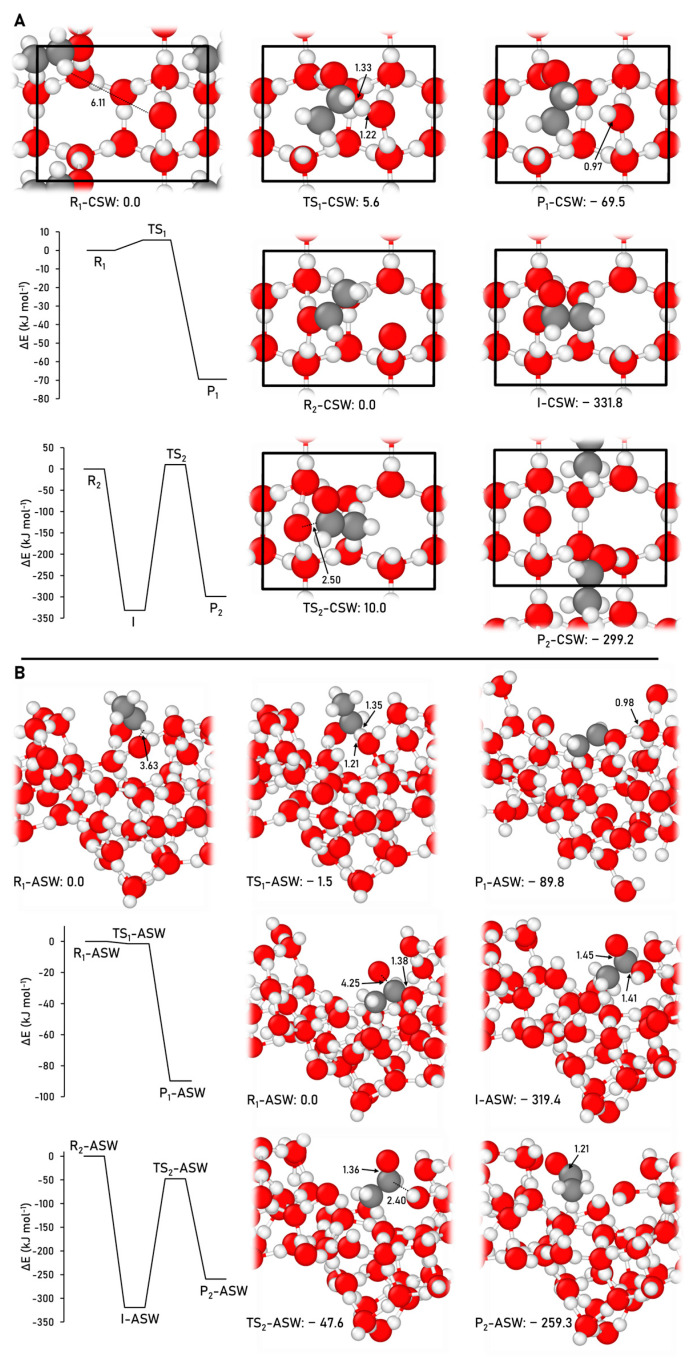
BHLYP-D3(BJ)//HF-3c ZPE-corrected PES and the corresponding optimized stationary points, alongside the relative energies, for the acetaldehyde formation on the CWS (**A**) and ASW (**B**) models. The unit cell is highlighted in black in the CWS model. Distances are in Å and relative potential energies in kJ mol^−1^. Color coding: white, H atoms; grey, C atoms; and red, O atoms.

**Figure 7 ijms-24-16824-f007:**
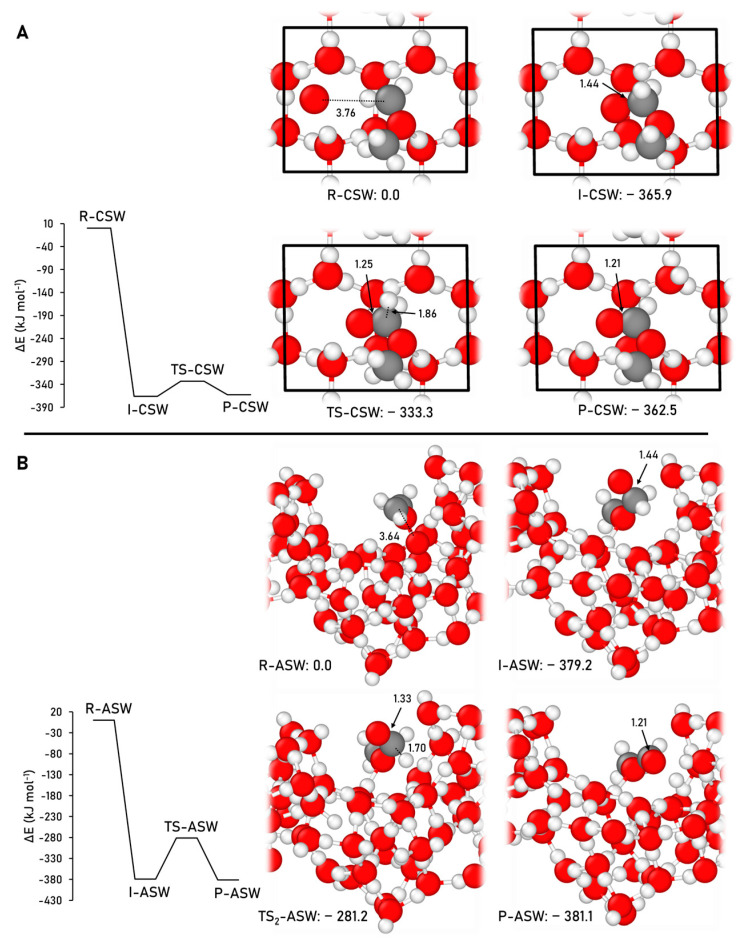
B3LYP-D3(BJ)//HF-3c ZPE-corrected PES, and the corresponding optimized stationary points, alongside the relative energies, for the methyl formate formation on the CWS (**A**) and ASW (**B**) models. The unit cell is highlighted in black in the CWS model. Distances are in Å and relative potential energies in kJ mol^−1^. Color coding: white, H atoms; grey, C atoms; and red, O atoms.

**Figure 8 ijms-24-16824-f008:**
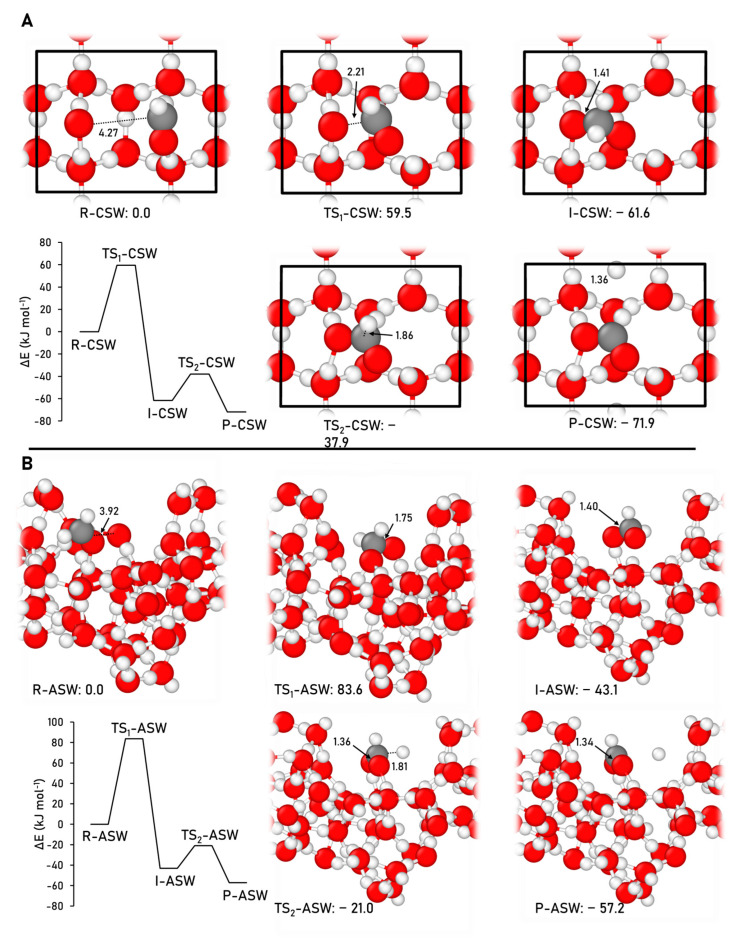
B3LYP-D3(BJ)//HF-3c ZPE-corrected PES and the corresponding optimized stationary points, alongside the relative energies, for the formic acid formation on the CWS (**A**) and ASW (**B**) models. The unit cell is highlighted in black in the CWS model. Distances are in Å and relative potential energies in kJ mol^−1^. Color coding: white, H atoms; grey, C atoms; and red, O atoms.

**Figure 9 ijms-24-16824-f009:**
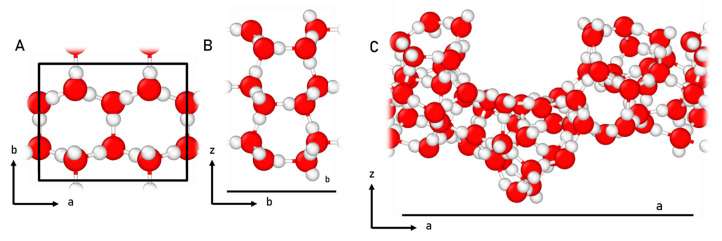
Top view (**A**) and lateral view (along the b lattice vector) (**B**) of the crystalline water surface model. Lateral view (along the a lattice vector) of the amorphous solid water surface (**C**). Unit cell highlighted in black. Color coding: white, H atoms; and red, O atoms.

**Table 1 ijms-24-16824-t001:** Computed ZPE-corrected energetics (in kJ mol^−1^) of the reactions simulated under different conditions (gas phase, and on the CWS and the ASW models). ∆E^‡^ stands for the intrinsic energy barriers and ∆E_X−Y_ for the relative energy between X and Y.

Reaction	Conditions	Energy Terms			
		∆E_1_^‡^	∆E_2_^‡^	∆E_I−R_	∆E_P−R_
	Gas phase	22.6	45.6	−34.4	−38.1
Reaction (1)	CWS	63.7	35.6	−41.6	−59.8
	ASW	51.7	38.4	−36.7	−45.0
		∆E^‡^	∆E_P1−R1_		
	Gas phase	18.1	−95.0		
Reaction (2)	CWS	5.6	−69.5		
	ASW	Barrierless	−89.8		
		∆E_1_^‡^	∆E_2_^‡^	∆E_I−R2_	∆E_P2−R2_
	Gas phase	Barrierless	72.8	−364.9	−312.0
Reaction (3)	CWS	Barrierless	341.8	−331.8	−299.2
	ASW	Barrierless	271.8	−319.4	−259.3
		∆E_1_^‡^	∆E_2_^‡^	∆E_I−R_	∆E_P−R_
	Gas phase	Barrierless	49.0	−381.6	−374.9
Reaction (4)	CWS	Barrierless	32.6	−365.9	−362.5
	ASW	Barrierless	98.0	−379.2	−381.1
		∆E_1_^‡^	∆E_2_^‡^	∆E_I−R_	∆E_P−R_
	Gas phase	33.1	59.2	−79.6	−63.1
Reaction (5)	CWS	59.5	23.7	−61.6	−71.9
	ASW	83.6	22.1	−43.1	−57.2

## Data Availability

Data are contained within the article and in Appendix A.

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
