# Peer review of "Gas-Phase vs. Grain-Surface Formation of Interstellar Complex Organic Molecules: A Comprehensive Quantum-Chemical Study"

_ijms, 2023, doi:10.3390/ijms242316824_

Round 1
Reviewer 1 Report
Comments and Suggestions for Authors
Look my file in attachment

Fine! Minor changes are required.
Author Response
See attached file for the answers.

Reviewer 2 Report
Comments and Suggestions for Authors
The paper by Martinez-Bachs and Rimola presents a computational quantum-chemical study on possible formation pathways of four interstellar organic molecules, comparing the gas phase and ice grain surface mechanisms. The conclusions are that (rather non-surprisingly) the ice surface modifies the energies of complexes and energy barriers, therefore the gas phase mechanism can not be effective on grain surfaces. The results are interesting and I would recommend acceptance of the paper after some issues are answered.
1. The benchmark calculations are unclear: the Authors take the geometries optimized in an arbitrarily chosen functional X (Table A1), calculate single point CCSD(T) energies, compare results with other functionals and then conclude that the CCSD(T) energies at the X-optimized geometries reproduce best the energies calculated in the functional X. This does not seem convincing. Will the “optimal” functional remain the same if the CCSD(T) energies are calculated at the geometries obtained in the functional Y, not X?
The DFT corrections are needed for the HF-3c data, therefore a more clear benchmark would be geometry optimization at the HF level with subsequent CCSD(T) energy calculations. Then the DFT functional could be selected on the basis of the best agreement of DFT//HF to CCSD(T)//HF results. But this is a comment; I do not suggest recalculate the data.
2. The energy unit J mol1 shown in Table A1 is obviously wrong.
3. Why the CCSD(T)/DFT energies shown in Figs. 2-5 differ from the values in Table A1? Because of the ZPE correction?
4. I understand that the analysis of the stationary points of PES was performed and the ZPE corrections were calculated at the HF-3c step of the DFT//HF-3c procedure. This should be explicitly stated.
5. Why was the amorphous water aggregate made periodic? There is no physical reason for it. Is this a requirement of the CRYSTAL program?
6. How the optimal adsorption site at the ASW model was found? The results obtained for the reactions will depend on the local geometry of ice surface at the adsorption site. Did the authors check if the energetics of a reaction will change at the other site? A related question is the geometry of the amorphous water aggregate – one can imagine a plethora of possible structures and they may lead to different energetic features of the reactions studied here. This issue should be discussed.
Author Response
See attached file for the answers.

Round 2
Reviewer 1 Report
Comments and Suggestions for Authors
The authors answered to all rises question and their work deserves to be published.
Comments on the Quality of English LanguageFine.